# Neural Scene Flow Prior

**Xueqian Li**[*1,2]     **Jhony Kaesemodel Pontes**[1]     **Simon Lucey**[2]
[1]Argo AI      [2]The University of Adelaide

## Abstract

Before the deep learning revolution, many perception algorithms were based on runtime optimization in conjunction with a strong prior/regularization penalty. A prime example of this in computer vision is optical and scene flow. Supervised learning has largely displaced the need for explicit regularization. Instead, they rely on large amounts of labeled data to capture prior statistics, which are not always readily available for many problems. Although optimization is employed to learn the neural network, the weights of this network are frozen at runtime. As a result, these learning solutions are domain-specific and do not generalize well to other statistically different scenarios. This paper revisits the scene flow problem that relies predominantly on runtime optimization and strong regularization. A central innovation here is the inclusion of a neural scene flow prior, which uses the architecture of neural networks as a new type of implicit regularizer. Unlike learning-based scene flow methods, optimization occurs at runtime, and our approach needs no offline datasets—making it ideal for deployment in new environments such as autonomous driving. We show that an architecture based exclusively on multilayer perceptrons (MLPs) can be used as a scene flow prior. Our method attains competitive—if not better—results on scene flow benchmarks. Also, our neural prior's implicit and continuous scene flow representation allows us to estimate dense long-term correspondences across a sequence of point clouds. The dense motion information is represented by scene flow fields where points can be propagated through time by integrating motion vectors. We demonstrate such a capability by accumulating a sequence of lidar point clouds.

## 1   Introduction

State-of-the-art results have recently been achieved by learning-based models [17, 30, 47, 60, 68] for the scene flow problem—the task of estimating 3D motion fields from dynamic scenes. However, such models heavily rely on large-scale data to capture prior knowledge, which is not always readily available. Scene flow annotations are expensive, and most methods train on synthetic and unrealistic scenarios to fine-tune on small real datasets.

Poor generalization to unseen, out-of-the-distribution inputs is another problem. Prior information is generally limited to the statistics of the data used for training. Real-world applications such as autonomous driving require robust solutions to low-level vision tasks such as depth, optical, and scene flow estimation that work in statistically different scenarios. Inspired by recent innovations that make use of coordinate-based networks (*i.e.*, pixels or 3D positions as inputs) [9, 36, 37, 39, 56] for 3D modeling and rendering, we investigate the use of such networks to regularize the scene flow problem without any learning directly from point clouds. Optimization happens at runtime, and instead of learning a prior from data, the network structure itself captures the prior information. It is not limited to the statistics of a specific dataset.

---

[*]Research done during internship at Argo AI. Corresponding e-mail: xueqian.li@adelaide.edu.au.

Optimizing neural networks at execution time is not new. Ulyanov *et al.* [63] showed that a randomly initialized convolutional network could be used as a handcrafted prior for standard inverse problems such as image denoising, super-resolution, and inpainting. Ding and Feng [12] proposed a runtime optimization method (DeepMapping) for rigid pose estimation using deep neural networks. Although such deep image priors, deep mapping, and coordinate-based networks for neural scene representations have been successfully applied for inverse problems, rendering, and rigid registration, none has yet investigated (to the best of our knowledge) the use of network-based priors for regularizing scene flow directly from point clouds.

Our proposed neural prior is based on a simple multilayer perceptron (MLP) architecture, and we show it is powerful enough to regularize scene flow given two point clouds implicitly. The input to the network is 3D points, and the output is a regularized scene flow.

Our neural prior allows for a continuous scene flow representation instead of discrete such as in graph Laplacian-based priors, *e.g.*, [45]. We show how the flow fields captured by our neural

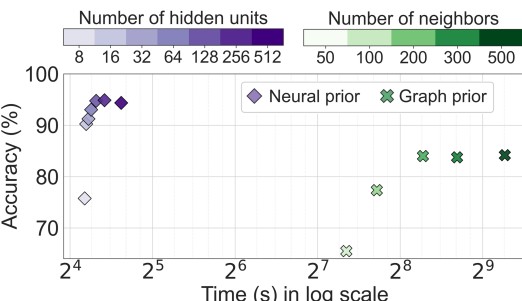

Figure 1: Our neural scene flow prior method achieved higher accuracy while being ∼10× faster than the recent runtime optimization method graph prior [45]. The evaluation was on the KITTI Scene Flow test set, where each point cloud size varies from 14k to 68k points. In our method, we fixed the number of hidden layers in the MLP to 4 and varied the number of hidden units. In the graph prior method, we varied the number of neighbors to create the graph. Accuracy uses the $Acc_5$ metric as defined in the experiments section. Learning-based methods might still be 10×–100× faster than the runtime optimization methods, but they still lack generalization and have memory issues when dealing with large point clouds—with tens of thousands of points.

prior can be employed to estimate long-term correspondences across a sequence of point clouds. The continuous scene flow allows for better integration of motions across time.

Our results are promising and competitive to supervised [30], self-supervised [38, 68], and non-learning methods [1, 45] (see Table 1). Our method also scales to real-world point clouds with tens of thousands of points while achieving better accuracy and time complexity than recent runtime optimization methods (see Fig. 1).

## 2  Related work

**Non-learning-based scene flow**    Scene flow is the uplift from the optical flow, which is proposed by Vedula *et al.* [64] as the non-rigid motion field in the 3D space. The authors proposed the optimization-based scene flow estimation using image sequences to infer reconstruction knowledge of the flow in 3D surfaces. Successive RGB/RGB-D image-based work [4, 18–21, 27, 43, 44, 50] used probability-based estimation, coarse-to-fine techniques, 6-DoF parameterization, or object segmentation, *etc.*, to improve accuracy and computation time. Although image-based scene flow methods are widely used, direct estimation of the scene flow from the point cloud is still possible through non-rigid registration methods, such as [1, 10, 26, 42]. In this paper, we focus on point cloud-based scene flow estimation.

**Learning-based scene flow**    Image-based learning methods [6, 51, 54, 60, 70] use convolution and data supervision to solve scene flow from monocular or RGB-D images with the available depth information. Other image-based methods [22, 23, 32, 52, 53] take care of extra occlusion cues in the large-scale autonomous driving scenes. Point-based learning methods have become more prevalent with the rapid development of point cloud feature learning [28, 48, 49, 65, 67]. FlowNet3D [30] is a seminal work that estimates scene flow using PointNet++ [49]. Successive work [17, 31, 47, 66] extends point-based learning methods using different feature extraction techniques. One obvious drawback of these supervised learning methods is the demand for sufficient ground truth labels. Besides, supervised methods lack generalizability while eventually only fitting domain-specific data. Self-supervised methods [25, 38, 61, 68], on the other hand, replaced the loss between the prediction and the ground truth flow with a point distance loss to use the point cloud itself as supervision.

Self-supervision can adapt to different datasets and maintain certain generalizability. Nonetheless, massive training data are still required for sufficient learning.

**The graph Laplacian method**   Graph Laplacian [2] is widely used to smooth the surface in mesh processing [5, 14, 57, 58], point cloud denoising [11, 71], *etc*. Here we talk about the recent scene flow estimation using Graph Laplacian [45]. The method explicitly constructed a graph of the point cloud to constrain the non-rigid scene flow as rigid within a specific range. While as a dataless runtime optimization, the method is heavily affected by the hyperparameters of the graph and loses scalability when the point cloud becomes larger or the neighbors in the graph grow.

**Deep neural prior, implicit functions, and neural rendering**   Although large-scale data helps in feature representation [59], end-to-end learning still requires high computation capacity and readily available training datasets. Instead, Ulyanov *et al.* [63] proposed a new style of optimization that uses a convolutional neural network to infer prior knowledge from the network architecture. One broader interest is to extend the idea of the network being an image function to the 3D shape modeling, and implicitly represent the continuous shape as level sets of neural networks. By directly mapping the 3D input to binary occupancy nets [9, 36], or signed distance functions [3, 39, 55], it is powerful to model the 3D geometries in a continuous space using the coordinate-based network. The following Scene Representation Networks [56] takes advantage of the coordinate-based network and renders view synthetic images. Mildenhall *et al.* proposed a seminal work NeRF [37], which is a novel way to do neural volume rendering using both point positions and viewing directions based on the coordinate-based network. Dynamic scene synthesis work [13, 16, 29, 40, 46, 62, 69] follows the NeRF framework, and integrates the motions to generate dynamic scenes. Some of the work [16, 29] use scene flow to further constrain or segment dynamic scenes. An interesting work that solves for the rigid alignment between point clouds using runtime optimization is DeepMapping [12]. However, this work only deals with rigid motion, and the network architecture is more complex than coordinate-based networks. In this work, we are interested in coordinate-based networks to address the large-scale, real-world scene flow problem.

## 3   Approach

**Problem definition**   Let $\mathcal{S}_1$ and $\mathcal{S}_2$ be two 3D point clouds sampled from a dynamic scene at time $t$-1 and $t$. The number of points in each point cloud, $|\mathcal{S}_1|$ and $|\mathcal{S}_2|$, are typically different and not in correspondence. A 3D point $\mathbf{p} \in \mathcal{S}_1$ moving from time $t$-1 to time $t$ can be modeled by a translational vector (or flow vector) $\mathbf{f} \in \mathbb{R}^3$, where $\mathbf{p}' = \mathbf{p} + \mathbf{f}$. The collection of flow vectors for all 3D points is the scene flow $\mathcal{F} = \{\mathbf{f}_i\}_{i=1}^{|\mathcal{S}_1|}$.

**Optimization**   We want to optimize for a scene flow $\mathcal{F}$ that minimizes the distance between the two point clouds, $\mathcal{S}_1$ and $\mathcal{S}_2$. Given the non-rigidity assumption of the scene, the optimization is inherently unconstrained. Thus, a regularization term C is necessary to constrain the motion field. We therefore solve for scene flow as

$$\mathcal{F}^* = \underset{\mathcal{F}}{\arg\min} \sum_{\mathbf{p} \in \mathcal{S}_1} D\left(\mathbf{p} + \mathbf{f}, \mathcal{S}_2\right) + \lambda C, \tag{1}$$

where D is a function to compute the distance from the perturbed point $\mathbf{p}$ by the flow vector $\mathbf{f}$ to its closest neighbor in $\mathcal{S}_2$. C is a regularizer (*e.g.*, Laplacian regularizer), and $\lambda$ is a weighting factor for the regularizer. In this paper, we want to investigate using a neural prior to regularize the scene flow.

### 3.1   Neural scene flow prior

Learning-based scene flow methods learn scene flow priors from a large number of examples. As in Deep Image Prior [63], we want to investigate if the structure of a neural network by itself is sufficient to capture a scene flow prior without any learning.

Here we use a neural network as an implicit regularizer. The parameters are optimized as

$$\boldsymbol{\Theta}^* = \underset{\boldsymbol{\Theta}}{\arg\min} \sum_{\mathbf{p} \in \mathcal{S}_1} D\left(\mathbf{p} + g\left(\mathbf{p}; \boldsymbol{\Theta}\right), \mathcal{S}_2\right), \tag{2}$$

where $g$ is a neural network parameterized by $\Theta$ to regularize the scene flow $\mathcal{F}$. The input to $g$ is $\mathbf{p}$, which is the point to be disturbed by the flow. The output of $g$ is $\mathbf{f}$ and thus $\mathbf{f}^* = g(\mathbf{p}; \Theta^*)$. For the distance function D, we define it as

$$\mathrm{D}(\mathbf{p}, \mathcal{S}) = \min_{\mathbf{x} \in \mathcal{S}} \|\mathbf{p} - \mathbf{x}\|_2^2. \tag{3}$$

In practice, we use it bidirectionally for both point sets, which is equivalent to Chamfer distance [15].

The objective in Eq. (2), is for the forward scene flow $\mathcal{F}$, which is the one we are interested in. However, it has been shown in [30, 38], that a cycle consistency regularizer encourages better scene flow estimations. The extra regularizer simply enforces the backward flow to be similar to the forward flow, $\mathcal{F}_{bwd} \approx \mathcal{F}$. The optimal backward flow is defined as $\mathbf{f}_{bwd}^* = g(\mathbf{p}'; \Theta_{bwd}^*)$, where $\mathbf{p}'$ is the shifted point by the forward flow as $\mathbf{p} + \mathbf{f}$. Note that the network $g$ is the same but with different parameters, $\Theta_{bwd}$. Using the backward flow as an additional constraint, the optimal network weights are solved as

$$\Theta^*, \Theta_{bwd}^* = \arg\min_{\Theta, \Theta_{bwd}} \sum_{\mathbf{p} \in \mathcal{S}_1} \mathrm{D}(\mathbf{p} + g(\mathbf{p}; \Theta), \mathcal{S}_2) + \sum_{\mathbf{p}' \in \mathcal{S}_1'} \mathrm{D}(\mathbf{p}' + g(\mathbf{p}'; \Theta_{bwd}), \mathcal{S}_1), \tag{4}$$

where $\mathcal{S}_1'$ is the shifted $\mathcal{S}_1$ by the forward flow, i.e., $\mathcal{S}_1' = \mathcal{S}_1 + \mathcal{F}$. Please find more details in the supplementary material. For the network $g$, we use MLPs with ReLU activations. The objective function in Eq. (4) can be optimized by gradient descent techniques using off-the-shelf frameworks with automatic differentiation. We show in the experiments section how the architecture of the neural prior affects performance by varying the number of hidden layers and units.

**Why use a neural scene flow prior?**  Deep learning relies on massive amounts of data and computational resources to capture prior statistics. Although learning methods have achieved impressive results in most tasks, they still struggle when deployed in environments where the statistics are different from those captured during learning. Our intuition is that a neural prior acts as a strong implicit regularizer that constrains dynamic motion fields to be as smooth as possible. A neural scene flow prior also scales to large scenes while achieving high-fidelity results at a low computational cost. Our proposed method with 8 hidden layers and 128 hidden units has about 116k parameters. FlowNet3D [30], for example, has about 1.2M parameters. Our method has $\sim$10$\times$ fewer parameters than the state-of-the-art supervised methods while achieving competitive, if not better, results. Lastly, our deep scene flow prior captures a continuous flow field that allows us to perform better scene flow interpolation across a sequence of point clouds.

## 4   Experiments

We evaluated the performance (accuracy, generalizability, and computational cost) of our neural prior for scene flow on synthetic and real-world datasets. We performed experiments on different neural network settings and analyzed the performance of the neural prior to regularizing scene flow. Remarkably, we show that a simple MLP-based prior to regularize scene flow is enough to achieve competitive results to the state-of-the-art scene flow methods.

**Datasets**  We used four scene flow datasets: **1) FlyingThings3D** [33] which is an extensive collection of randomly moving synthetic objects. We used the preprocessed data from [30]; **2) KITTI** [34, 35] which has real-world self-driving scenes. We used the subset released by [30]; **3) Argoverse** [8] and **4) nuScenes** [7] are two large-scale autonomous driving datasets with challenging dynamic scenes. However, there are no official scene flow annotations. We followed the data processing method in [45] to collect pseudo-ground-truth scene flow. Ground points were removed from lidar point clouds as in [30] (please refer to the supplementary material for more details).

**Metrics**  We employed the widely used metrics as in [30, 38, 45, 68] to evaluate our method, which are: $\mathcal{E}$ to denote the end-point error (EPE), which is the mean absolute distance of two point clouds; $Acc_5$ to denote the accuracy in percentage of estimated flows when $\mathcal{E} < 0.05$m or $\mathcal{E}' < 5\%$, where $\mathcal{E}'$ is the relative error; $Acc_{10}$ denotes the percentage of estimated flows where $\mathcal{E} < 0.1$m or $\mathcal{E}' < 10\%$; and $\theta_\epsilon$ which is the mean angle error between the estimated and ground-truth scene flows.

**Implementation details** We defined our neural prior for scene flow as a simple coordinate-based MLP architecture with 8 hidden layers, a fixed length of 128 for the hidden units, Rectified Linear Unit (ReLU) activation and shared weights across points. The network input is the 3D point cloud $\mathbf{P}_{t\text{-}1}$, and the output is the scene flow $\mathbf{F}$. We used PyTorch [41] for the implementation and optimized the objective function with Adam [24]. The weights were randomly initialized. We set a fixed learning rate of $8e{-}3$ and run the optimization for 5k iterations with early stopping on the loss. For our settings and datasets, we found the optimization to mostly converge in less than 1k iterations. All experiments were run on a machine with an NVIDIA Quadro P5000 GPU and a 16 Intel(R) Xeon(R) W-2145 CPU @ 3.70GHz.

**Training setup for the learning-based methods** We used the publicly available implementation of the learning-based methods to perform our experiments. The training settings for each method are: **FlowNet3D** and full-supervised **PointPWC-Net**: trained on FlyingThings3D with supervision; and self-supervised **Just Go with the Flow** and **PointPWC-Net**: trained on FlyingThings3D with supervision and fine-tuned on domain-matched datasets with self-supervision (*i.e.*, fine-tuned and tested on statistically similar data, KITTI, nuScenes, Argoverse respectively). Note that FlowNet3D and full-supervised PointPWC-Net was only trained on the synthetic FlyingThings3D to demonstrate the poor generalizability of learning-based methods to other domains. Just Go with the Flow and self-supervised PointPWC-Net were trained using self-supervision, and although they do not require ground-truth annotations, they still require large-scale datasets for training to achieve competitive performance. Note that these self-supervised methods were both pretrained on fully-labeled FlyingThings3D to provide adequate full supervision.

**Optimization setup for the non-learning methods** Non-rigid ICP [1] was originally proposed for mesh registration. We adapted it for point cloud registration. A graph prior was recently proposed in [45] to optimize scene flow from point clouds. We implemented the method using the hyperparameters defined by the authors. The weight for the graph prior term is set to 10, the number of neighbors $k$ to build the $k$-NN graph, if not explicitly specified, is set to 50, the learning rate to 0.1, and the number of iterations to 1.5k.

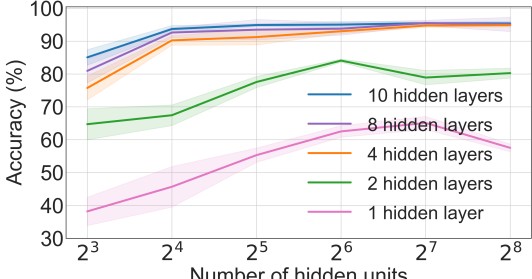

Figure 2: We analyzed the performance of our method on the KITTI test set when varying the number of hidden layers and hidden units of the MLP architecture. Accuracy is the $Acc_5$ metric.

### 4.1 Choosing the neural prior architecture

Fig. 2 shows how the performance of our method is affected when varying the neural prior MLP architecture: number of hidden layers and hidden units. The experiments were performed on the KITTI test set and all points included (*i.e.*, without point sampling). The average number of points for the KITTI dataset is about 30k. We ran our method five times with different random seeds to include the uncertainty levels in the plot.

Overall, the performance of our method improved as we increased the number of hidden layers and hidden units. For small numbers of hidden layers (*e.g.*, 1 and 2), the performance deteriorated when the number of hidden units is large, around 128 and 256 (or $2^7$ and $2^8$). We chose the MLP architecture with the best performance with relatively low computation time for our following experiments: 8 hidden layers and 128 ($2^7$) hidden units.

### 4.2 Comparing to other methods

Table 1 shows how our method stands against other state-of-the-art methods on different datasets and metrics. We set the number of points to 2,048, which follows the experimental protocols as in FlowNet3D [30] and Graph prior [45]. We ran the experiments 5 times to report uncertainties for runtime optimization methods (*i.e.*, our method and the graph prior method [45]) with uncertainties for each run. The learning-based methods and non-rigid ICP are deterministic during runtime.

Our method achieved better performance in most datasets and metrics. We considered the well-known Non-rigid ICP as a baseline for the non-learning methods (●). Our method outperformed the recent

Table 1: Performance of our method and others across different datasets and metrics. ● are supervised methods trained on the synthetic FlyingThings3D dataset. ● are self-supervised methods trained on FlyingThings3D with supervision and fine-tuned with self-supervision on matched datasets. ● are non-learning methods that do not rely on training data. All experiments were run with 2,048 points. ↑ means larger values are better while ↓ means smaller values are better. We did no report standard deviations smaller than $1e-2$.

| | FlyingThings3D [33] Train: 19,967 samples, Test: 2,000 samples | | | | nuScenes Scene Flow [7] Train: 1,513 samples, Test: 310 samples | | | |
|---|---|---|---|---|---|---|---|---|
| | $\mathcal{E}(m)\downarrow$ | $Acc_5(\%)\uparrow$ | $Acc_{10}(\%)\uparrow$ | $\theta_\epsilon(rad)\downarrow$ | $\mathcal{E}(m)\downarrow$ | $Acc_5(\%)\uparrow$ | $Acc_{10}(\%)\uparrow$ | $\theta_\epsilon(rad)\downarrow$ |
| ● FlowNet3D [30] | 0.134 | 22.64 | 54.17 | 0.305 | 0.505 | 2.12 | 10.81 | 0.620 |
| ● PointPWC-Net [68] | **0.121** | **29.09** | **61.70** | **0.229** | 0.442 | 7.64 | 22.32 | 0.497 |
| ● Just Go with the Flow [38] | — | | | | 0.625 | 6.09 | 0.139 | 0.432 |
| ● PointPWC-Net [68] | — | | | | 0.431 | 6.87 | 22.42 | 0.406 |
| ● Non-rigid ICP [1] | 0.339 | 14.05 | 35.68 | 0.480 | 0.402 | 6.99 | 21.01 | 0.492 |
| ● Graph prior [45] | 0.255 | 16.56±0.02 | 42.05±0.02 | 0.362 | 0.289 | 20.12±0.01 | 43.54±0.02 | 0.337 |
| ● Ours | 0.234 | 19.16±0.23 | 46.74±0.46 | 0.341 | **0.175±0.01** | **35.18±1.32** | **63.45±0.46** | **0.279±0.04** |
| | KITTI Scene Flow [34, 35] Train: 100 samples, Test: 50 samples | | | | Argoverse Scene Flow [8] Train: 2,691 samples, Test: 212 samples | | | |
| | $\mathcal{E}(m)\downarrow$ | $Acc_5(\%)\uparrow$ | $Acc_{10}(\%)\uparrow$ | $\theta_\epsilon(rad)\downarrow$ | $\mathcal{E}(m)\downarrow$ | $Acc_5(\%)\uparrow$ | $Acc_{10}(\%)\uparrow$ | $\theta_\epsilon(rad)\downarrow$ |
| ● FlowNet3D [30] | 0.199 | 10.44 | 38.89 | 0.386 | 0.455 | 1.34 | 6.12 | 0.736 |
| ● PointPWC-Net [68] | 0.142 | 29.91 | 59.83 | 0.239 | 0.405 | 8.25 | 25.47 | 0.674 |
| ● Just Go with the Flow [38] | 0.218 | 10.17 | 34.38 | 0.254 | 0.542 | 8.80 | 20.28 | 0.715 |
| ● PointPWC-Net [68] | 0.177 | 13.29 | 42.15 | 0.272 | 0.409 | 9.79 | 29.31 | 0.643 |
| ● Non-rigid ICP [1] | 0.338 | 22.06 | 43.03 | 0.460 | 0.461 | 4.27 | 13.90 | 0.741 |
| ● Graph prior [45] | 0.099 | 63.60±0.09 | 81.18±0.08 | 0.176 | 0.257 | 25.24±0.04 | 47.60±0.02 | 0.467 |
| ● Ours | **0.050±0.01** | **81.68±2.00** | **93.19±1.30** | **0.133±0.01** | **0.159±0.01** | **38.43±0.48** | **63.08±0.59** | **0.374±0.01** |

graph prior method by a large margin. The supervised FlowNet3D and PointPWC-Net (using full-supervision loss) methods (●) had better performance on FlyingThings3D because they were trained on it with supervision. If the dataset is out-of-the-distribution, these supervised methods produced unreliable results. The self-supervised methods (●), Just Go with the Flow and PointPWC-Net (using self-supervision loss), despite not being exposed to ground-truth labels during training, still generated better results than supervised methods—showing that self-supervision is an important direction for scene flow estimation. Still, it is remarkable that with a simple MLP regularizer and an optimization framework, our method can robustly estimate scene flow from point clouds with great accuracy.

Our PointPWC-Net results were different from those reported in the PointPWC-Net paper. The reasons are: In the official PointPWC-Net implementation, there is a threshold to limit the lidar point cloud within 35 meters of distance from the center. In our experiments, we used all points available in all ranges (up to 85 m). Lidar point clouds get sparser as the distance increases, making it challenging to estimate scene flow in far ranges and sparse regions. Nevertheless, we did not shy away from this fact in our experiments. Also, in the original PointPWC-Net experiments, the authors used 8,192 points. In ours, we used 2,048 points. Naturally, there exists a performance gap between our reported results and theirs. We decided to use 2,048 points to follow the experiment protocols proposed in FlowNet3D [30] and graph prior [45] to facilitate comparisons across different datasets and models. Although simple and tested on sparse point clouds (2,048 points), our method achieved impressive results on different datasets. Please find further details and additional experiments in the supplementary material.

### 4.3 Estimating scene flow from large point clouds with high density

Real-world point clouds collected from depth sensors such as lidar typically have tens of thousands of points. We evaluated the performance of our method on large point clouds and compared it against the graph prior method. The KITTI and Argoverse Scene Flow datasets were used, given that both have large point clouds with high density.

Fig. 3 shows the performance on the KITTI Scene Flow dataset, in terms of accuracy and computational time, of our method and the graph prior method when varying the number of points. Our method's accuracy ($Acc_5$) increased as the number of points grew until around 20k points and then saturated, while the computational time slowly increased. In contrast, the graph prior achieved lower accuracy and dramatic growth in computation. The computational complexity of our MLP-based prior grows linearly in the number of points, $\mathcal{O}(n)$, while the graph prior grows quadratically

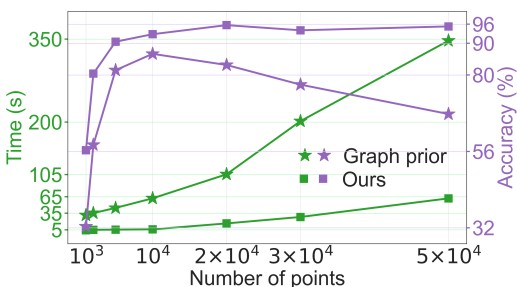

Figure 3: Performance of our neural prior and the graph prior [45] when varying the number of points. Our method achieved higher accuracy ($Acc_5$) and better time complexity. Results were averaged over the KITTI Scene Flow dataset.

Table 2: Performance of our neural prior and the graph prior [45] when using all points available. Our method achieved better performance on all metrics by a margin while being $\sim5\times$ faster if $k{=}50$ and $\sim10\times$ faster if $k{=}200$ (for KITTI).

| | $\mathcal{E}\downarrow$ (m) | $Acc_5\uparrow$ (%) | $Acc_{10}\uparrow$ (%) | $\theta_\epsilon\downarrow$ (rad) | Time$\downarrow$ (s) |
|---|---|---|---|---|---|
| KITTI Scene Flow *Average number of points: $\sim$30k* | | | | | |
| Graph prior ($k$=50) | 0.225 | 65.50 | 70.32 | 0.277 | 162.97 |
| Graph prior ($k$=200) | 0.082 | 84.00 | 88.45 | 0.141 | 310.12 |
| Ours | **0.025** | **95.68** | **98.00** | **0.085** | **38.33** |
| Argoverse Scene Flow *Average number of points: $\sim$50k* | | | | | |
| | $\mathcal{E}\downarrow$ (m) | $Acc_5\uparrow$ (%) | $Acc_{10}\uparrow$ (%) | $\theta_\epsilon\downarrow$ (rad) | Time$\downarrow$ (s) |
| Graph prior ($k$=50) | 0.249 | 46.92 | 61.72 | 0.494 | 410.21 |
| Ours | **0.043** | **86.04** | **94.07** | **0.244** | **84.46** |

in the number of points, $\mathcal{O}(n^2)$. The graph prior relies on the construction of a graph Laplacian matrix to use as a regularizer (*i.e.*, $\mathbf{L} \in \mathbb{R}^{n \times n}$, where $n$ is the number of points).

The accuracy of the graph prior method degraded after 10k points. The graph regularizer needs more than 5k iterations for higher density point clouds to converge to a reasonable solution or carefully tuned schedulers to accelerate its convergence. Moreover, the $k$-NN graph is built with 50 neighbors, and for higher density point clouds, larger graphs might be necessary for a better regularization.

Table 2 shows a quantitative comparison between our method and the graph prior method on the KITTI and Argoverse Scene Flow datasets when using all points. Our method achieved better performance on all metrics. We also reported results for the graph prior method when setting the number of neighbors, $k$, to 200. According to our results in Fig. 1, the scene flow accuracy saturated after $k$=200. Fig. 4 shows a qualitative example of a scene flow estimation using our method.

These results show that our method scales to large point clouds with high density, and gain a great improvement in performance with much denser point clouds. In contrast, training supervised/self-supervised models with high-density point clouds is not always practical due to high memory usage. Typically, such models are trained with up to 8k points.

**Performance and inference time trade-offs** Estimating scene flow with runtime optimization is usually slower than using learning-based methods ($10\times$–$100\times$ slower). The trade-off, however, depends on the application. If robustness/generalizability is not an issue but rather the inference time, our proposed objective can train a self-supervised model and act as a surrogate of our non-learning method but inheriting the faster inference time from the trained model. We show an example of lidar point cloud densification (Section 4.6) to generate denser point clouds that can be used for robotics applications such as offline mapping, creating denser depth maps, *etc*., that would not require real-time inference.

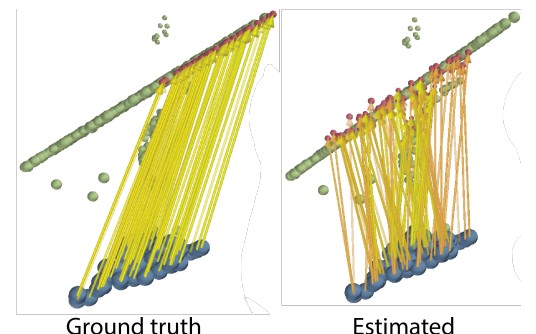

Ground truth        Estimated

Figure 5: Example of a failure case. Partial scene from FlyingThings3D. Our nearest-neighbor-based loss might fail when handling large missing parts, occlusions, and bad correspondences. Green points are the target, and red points are the shifted blue points by the estimated scene flow (yellow arrows).

### 4.4 Limitations

Although our method achieved better computational complexity than other non-learning-based methods, the inference time is still limiting for some applications that demand real-time inferences.

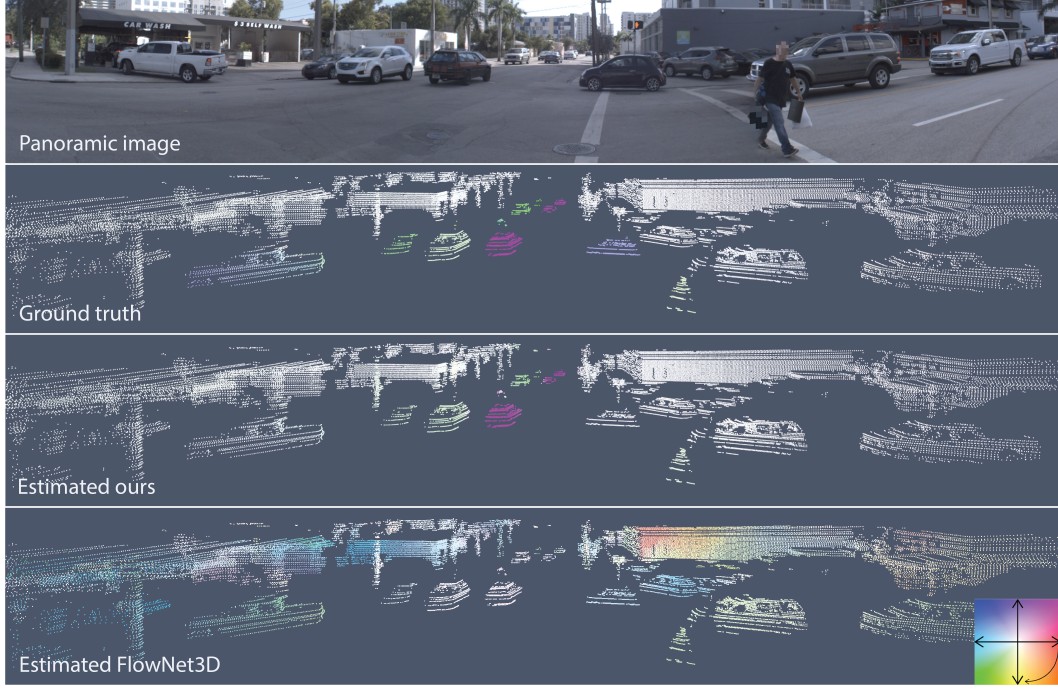

Figure 4: Qualitative example of a scene flow estimation using our proposed method. The complex and highly dynamic driving scene is from the Argoverse Scene Flow dataset. The scene flow estimated by our method is close to the ground truth. We also show a prediction using the supervised FlowNet3D method trained on FlyingThings3D and fine-tuned on the KITTI Scene Flow dataset. Note how the scene flow deviated from the ground truth when the inference was performed on an out-of-the-distribution sample. The scene flow color encodes the magnitude (color intensity) and direction (angle) of the flow vectors. For example, the purplish vehicles are heading northeast.

Another limitation is that the loss function we used relies on nearest neighbors, which might find bad correspondences due to partial point clouds and occlusions. Fig. 5 shows a failure case because of the nearest-neighbor-based distance loss. Few corresponding points due to missing parts in the scene might lead to incorrect flow estimations.

## 4.5   A continuous scene flow field

Our neural prior implicitly regularizes the scene flow through the coordinate-based MLP network. Thus, our method allows for a continuous scene flow representation instead of a discrete representation such as in graph-based priors.

An advantage of a continuous scene flow representation is that we can reuse the optimal network weights to estimate dense long-term correspondences across a sequence of point clouds (see Section 4.6). Fig. 6 shows how the estimated scene flow and the continuous flow field change as the optimization converges to a solution.

## 4.6   Application: scene flow integration

Here we demonstrate how to use our method to perform scene flow integration. Given a temporal sequence of point sets, $\{\mathcal{S}_0, \mathcal{S}_1, \mathcal{S}_2, \cdots, \mathcal{S}_M\}$, we first optimize for the pairwise scene flows, $\{\mathcal{F}_{0\to1}, \mathcal{F}_{1\to2}, \cdots, \mathcal{F}_{M\text{-}1\to M}\}$, using our proposed method with the optimal neural prior parameters, $\{\mathbf{\Theta}^*_{0\to1}, \mathbf{\Theta}^*_{1\to2}, \cdots, \mathbf{\Theta}^*_{M\text{-}1\to M}\}$, saved. Then, starting from $\mathbf{f}_{0\to1}$, we can integrate long-term flows using the classic Forward Euler method recursively for $m=1{:}M\text{-}1$ iterations as

$$\mathbf{f}_{0\to m+1} = \mathbf{f}_{0\to m} + g(\mathbf{p}_0 + \mathbf{f}_{0\to m}; \mathbf{\Theta}^*_{m\to m+1}). \tag{5}$$

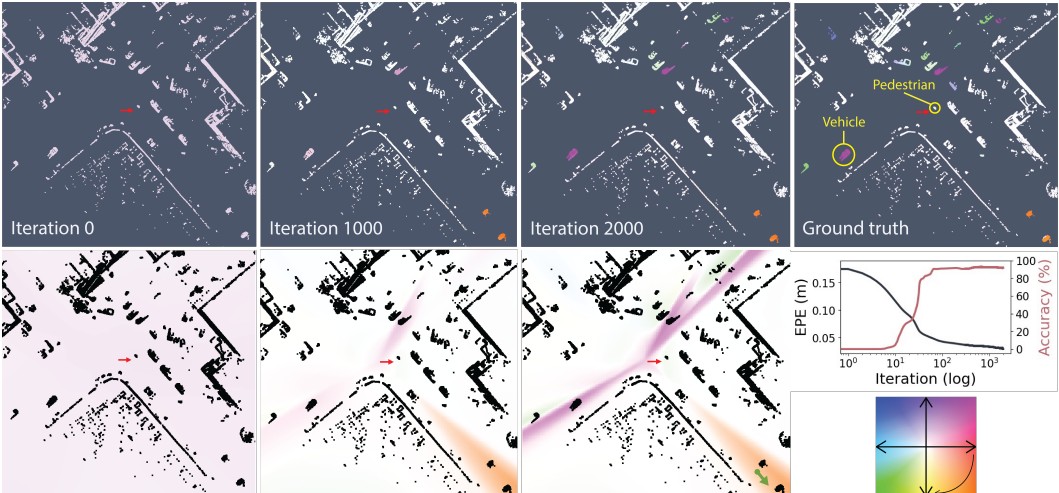

Figure 6: Example showing how the estimated scene flow and the continuous flow field (bottom) given by our neural prior change as the optimization converges to a solution. We show a top-view dynamic driving scene from Argoverse Scene Flow. The scene flow color encodes the magnitude (color intensity) and direction (angle) of the flow vectors. For example, the purplish vehicles are heading northeast. The red arrow shows the position and direction of travel of the autonomous vehicle, which is stopped, waiting for a pedestrian to cross the street. Note how the predicted scene flow is close to the ground truth at iteration 2k. At iteration 0, the scene flow is random, given the random initialization of the neural prior. Thus having very small magnitudes for the random directions. As the optimization went on, the flow fields became better constrained. A simple way to interpret the flow fields is to imagine sampling a point at any location in the continuous scene flow field to recover an estimated flow vector. For example, imagine sampling a point around the orange region in the flow field at iteration 2k (green arrow in the bottom right). The direction of the flow vector will be pointing southeast at a specific magnitude, similar to the vehicles in the orange region.

This gives the long-term scene flow $\mathcal{F}_{0 \to M} = \{(\mathbf{f}_{0 \to M})_i\}_{i=0}^{|\mathcal{S}_0|}$, from $\mathcal{S}_0 \to \mathcal{S}_M$. Note that we are not relying on discrete nearest-neighbor-based interpolations. Our neural scene flow prior is a continuous representation that naturally provides continuous scene flow estimations. Fig. 7 shows an example of an Argoverse scene where we applied such a technique to integrate 10 point clouds into a single frame to densify the point cloud.

## 5   Conclusion

We show that how hand-designed coordinate-based network architecture can serve as a new type of implicit regularizer in the runtime optimization for the scene flow problem. Our neural prior gets rid of the need for massive labeled/unlabeled training data while being scalable with dense point clouds. Additionally, since we infer prior knowledge from the network architectures instead of from data, our approach can generalize to out-of-the-distribution scenarios as compared to learning-based methods. The continuous flow representation also allows for flow integration across a long sequence that can be used in many robotics applications such as offline mapping. We believe this paper shows a promising direction for large-scale, real-world scene flow estimation without data supervision.

### Broader impact

Our proposed neural scene flow prior allows for estimating 3D motion fields from large-scale dynamic scenes without annotating massive data but preserving generalizability—making it useful for scenarios where motion prediction is required. Especially in computer vision and robotics communities, robust scene flow estimations is crucial. For example, autonomous vehicles need to predict future distribution of the surrounding objects to avoid catastrophe in dynamic environments; safe human-computer interaction is enabled with precise dynamic flow predictions. Our work also encourages further

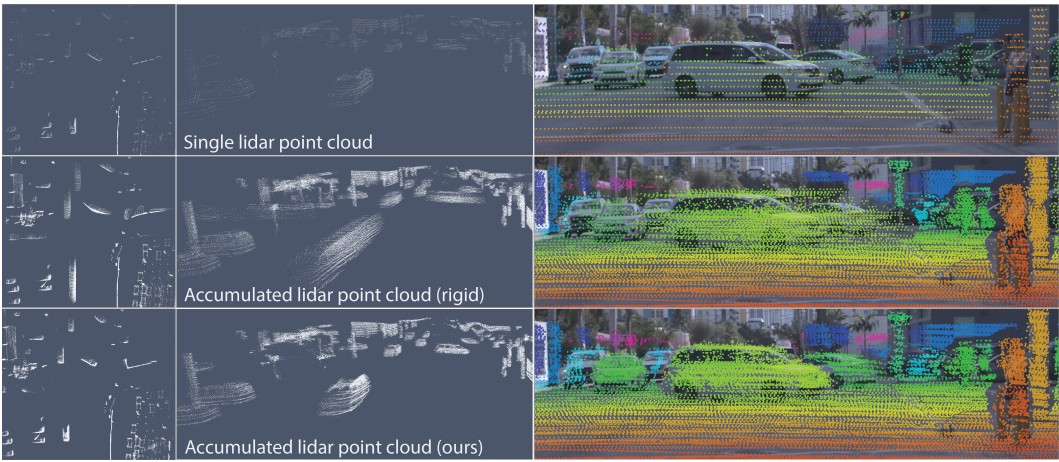

Figure 7: Example of a scene flow integration to densify an Argoverse lidar point cloud. The left and middle columns are a top and front view of the point cloud, respectively. The rightmost column shows the accumulated point cloud projected onto the image. Note the smearing effect on the dynamic objects when rigidly accumulating the point clouds (middle row). Accumulation using our neural prior nicely produced a denser point cloud while taking care of all dynamic objects in the scene. Here, rigid means that the point cloud accumulation was performed using a rigid registration method (*i.e.*, ICP) where rigid 6-DoF poses are used for the registrations.

exploration in the combination of the innovative coordinate-based networks and classical runtime optimization algorithm. This dataless approach offers an affordable solution for many industrial problems without adequate supervision (*e.g.*, autonomous driving).

However, as AI-based research, it could be misused by malicious groups for nefarious purposes. For example, the collected data might contain sensitive information that potentially invades privacy, and can be used for illegal data trading. Moreover, such research has the potential to be used in autonomous weapons and military drones. The potential evil use of our method needs attention and needs to be prevented. We hope to motivate the community to take full advantage of the innovations of this work to benefit society.

## Acknowledgments and Disclosure of Funding

The authors would like to thank Chen-Hsuan Lin for useful discussions through the project, review and help with section 3. We thank Haosen Xing for careful review of the entire manuscript and assistance in several parts of the paper, Jianqiao Zheng for helpful discussions. We thank all anonymous reviewers for their valuable comments and suggestions to make our paper stronger.

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
