# OpenReview forum: "Neural Scene Flow Prior"
_NeurIPS.cc/2021/Conference — NeurIPS 2021 Spotlight_

### Official Review · Reviewer_rKbV · 2021-07-14

**Rating:** 7
**Confidence:** 4

**Summary:**


This paper highlights the fact that deep learning has caused a paradigm shift in many computer vision applications from methods based on optimization with strong priors to ones that optimize  feed-forward neural networks in an offline process. The paper argues that these feed-forward models are bad at generalization and proposes to use the network in a radically different way. In essence, the network are used as a parametrization for a traditional optimization solution - allowing the network's inductive biases to be used as a “prior”. While using a network structure as a prior for optimization is not new (ala Deep Image Prior, DeepSDF, NeRF etc.), applying this type of parametrization to the problem of optical flow is new. The results are competitive with trained approaches and quite impressive for such a simple solution.

**Limitations And Societal Impact:**


The discussion of the potential negative societal impacts is limited to "as an AI-based research, it could be misused" - which I don't feel is satisfactory. Being a method for tracking people, objects and things accurately I think the authors should at least mention immediate ways in which it could be misused for nefarious purposed - e.g. invading privacy by tracking pedestrians across the city, or it's potential use in autonomous weapons and drones. I only mention these as the authors have described many potential positive impacts but I feel the discussion should be more balanced.

**Main Review:**


Overall this is a well-written and interesting paper. Please find my detailed comments below where I have labelled each point as either positive (+), mixed (+/-) or negative (-).

**Originality**:

(+) The method is new in the sense that a neural parameterization of this type has not been used before for scene flow estimation. This type of parameterization has multiple advantages such as the fact that the scene flow becomes a continuous field that can be sampled at any point in space which is a much more natural way to represent the flow field. In contrast traditional approaches based on variational methods would typically solve for the flow field only on 2D frames of projected points.

(+) In my view, the work ***is*** a novel combination of well-known techniques. Optimization-based approaches for scene flow have existed for a long time and quite sophisticated methods have been developed. Similarly, the use of the inductive biases of deep networks as a prior to guide optimization was first described in “deep image prior” and has been exploited in multiple subsequent works. However, the application of these to the problem of scene flow estimation is novel and the authors have done a very good job in describing how their approach bridges the different paradigms.

(+/-) Overall, the work is well-positioned with respect to previous contributions in the space (like Deep image prior, supervised methods, and traditional optimization approaches). However, there is one very related work that seems to be missing and that is DeepMapping [A]. Like the proposed approach, DeepMapping also aligns point clouds in an “unsupervised manner” (i.e. test time optimization) and uses a deep network for the parametrization for the pose. I do think there are clear differences between the proposed method and DeepMapping in the sense that the proposed method recovers a full, continuous flow field whereas DeepMapping only recovers the pose. This means that the proposed method can compute the flow for dynamic objects whereas DeepMapping is limited to rigid scenes. The proposed method also appears to be much simpler than DeepMapping in terms of its design and training. However, it would probably be good to reference and compare DeepMapping to the proposed approach (in terms of a description of the differences) as it probably is the closest existing method. It would also be interesting to see how DeepMapping performs compared to the proposed method on rigid scenes.

**Quality**:

(+) As far as I can tell, the submission is technically sound and there are no major errors in the method.
The claims related to the proposed approach (in terms of performance compared to existing methods) are well supported. The numerical experimentation is sufficient to allow the reader to place the method with respect to recent supervised approaches and the results do indicate that the method is very competitive with these approaches.

(+) The paper is relatively polished and complete in the sense that the idea is fully described and evaluated to an extent that allows the reader to see that the approach works and what problems it is suited for.  It would have been nice to see more demos on different types of data (currently only lidar point clouds from vehicles are shown) and I’m really curious how this would work on denser point clouds from RGB-D sensors in dynamic indoor scenes. The attached video clip in the supplementary showing the approach in-action is a bit difficult to understand as it’s just a screen grab of the python program running with no editing. However, I don’t think these issues detract significantly from the quality of the paper.

(-) The authors do a good job of highlighting the strengths of the method, but fall a bit short in describing the weaknesses. The main advantage of approaches that are only optimized at training time is of course their inference speed and this has been one of the major factors that has led to the widespread adoption of deep learning models. Whereas traditional state-of-the-art scene flow methods would take minutes or hours to process a single example, methods like FlowNet3D can do it in seconds. In essence, there exists a strong tradeoff between generalization and inference speed and I think it is quite important for the authors to make this clear in the paper. Inference time is currently only compared to GraphPrior in Figure 3, but I think inference times should be given for other methods like FlowNet3D as well (perhaps by adding them to Table 1).

**Clarity**:

(+) In general the submission is very clearly written, and makes a good argument for the neural prior and optimization-based approach that is proposed. The paper is well organized and contains all the details to replicate the approach (the authors have also provided the source code for reproducibility).

**Significance**:

(+) In my view the results are quite important for two reasons. It shows that a very simple optimizer (Adam) combined with an easy to implement loss function and neural prior can achieve very good results on the scene flow task - even beating supervised approaches in some cases. Secondly, it shows that a continuous representation is the best way to represent 3D flow fields (similar to what has been shown on other 3D tasks too - like for volume rendering in Nerf). Therefore I think these result could make a large impact in this research area too.

(+) Furthermore, I do think other researchers are very likely to be inspired by the approach and try to build on top of it to overcome its currently limitations (like inference speed and pushing the accuracy to maybe beat the current supervised methods). This is because the proposed method opens up many interesting possibilities for future research.

(+) Although the method does not advance the state of the art in terms of numerical performance, it absolutely presents a method that is better in terms of complexity and ease of implementation compared to previous work. In my view, the simplicity of the approach, and the advantages afforded by being able to reason about a full, continuous flow field are what really make this a great paper.

[A] Li Ding, Chen Feng, DeepMapping: Unsupervised Map Estimation From Multiple Point Clouds, CVPR 2019.

# Post-rebuttal and discussion:

After considering the authors' response and the other reviews, I am still positive about this paper and therefore give it a final Rating of "7: Good paper, accept". I think the idea of using a neural network parameterization for test-time optimization to solve for non-rigid flow is a very interesting contribution and promising area of research and the paper shows strong results to back this up. The authors have promised to include the missing reference and discussion about DeepMapping [A], as well as clarifying the limitations of the approach.


**Time Spent Reviewing:**

2

---

> ### Author Response · Authors · 2021-08-10
> **Response to Reviewer rKbV**
>
> We appreciate reviewer rKbV for the thoughtful comments. We are happy to know our idea is novel, simple, the paper is well written, and we feel motivated by the many other qualities the reviewer pointed out.
>
> We address the reviewer's concerns and questions below.
>
> **Q1: A very related work that seems to be missing is DeepMapping.**
>
> **A1:** We agree that there are fundamental differences between DeepMapping and our proposed work. DeepMapping solves for a rigid transformation while our method solves for a non-rigid transformation -- the collection of translational vectors or scene flow. Nevertheless, DeepMapping also investigated implicit representations using neural networks for runtime optimization and deserves to be cited and discussed in our paper.
>
> **Q2: Experiments on dynamic indoor point cloud obtained from RGB-D sensors.**
>
> **A2:** We agree. However, to the best of our knowledge, there is no real-world dynamic indoor dataset that provides point clouds and scene flow labels yet for a complete evaluation.
>
> **Q3: Did not fully discuss the weaknesses (*e.g*., the tradeoff between generalization and inference speed).**
>
> **A3:** We agree that we should further discuss the tradeoffs between generalization and inference speed. Similar to our reply to reviewer XKk4, our method is not yet competitive to learning-based methods *w.r.t* inference time. We have shown that our proposed scene flow objective function can be directly optimized during runtime to achieve robust results and generalize to other domains. Nevertheless, our method could also be explored to self-supervise the learning of scene flow from point clouds using, for example, an off-the-shelf scene flow model. The tradeoff will depend on the application. For example, suppose robustness/generalizability is not an issue but rather the inference time. In that case, our proposed objective function can train a self-supervised model and act as a surrogate of our non-learning method but with faster inference time.
>
> **Q4: The social impact discussion should be more balanced.**
>
> **A4:** Yes, we are generally biased to the positive side of our research, but we should take the time to reflect on the negative societal impacts. Therefore, we are happy to think about it and make the societal impact section more balanced.

---

> > ### Comment · Reviewer_rKbV · 2021-08-23
> > **Discussion and comparison to DeepMapping**
> >
> > Thank you for the response.
> >
> > Most of the concerns raised in my initial review have been addressed, however, I think the comparison to, and discussion of DeepMapping is rather important as it is probably the closest related approach. The proposed approach does account for non-rigid scenes, but most of the evaluation is done on static scenes so I don't see a problem with comparing the approaches in the evaluation. I expect the proposed approach would probably outperform DeepMapping but this can only be proved through an actual evaluation.
> >
> > I do believe that this issue needs to be addressed before the paper can be accepted.

---

> > > ### Author Response · Authors · 2021-08-25
> > > **Response to additional comments by reviewer rKbV**
> > >
> > > Thank you for the additional comments.
> > >
> > > We want to first remind the reviewer that our evaluations were performed on ***dynamic*** scenes with ***non-rigid*** moving objects.
> > > Previous methods, FlowNet3D, Graph Prior, PointPWC-Net, *etc*. have shown that scene flow methods outperform rigid methods when scenes are dynamic.
> > > We do not see the necessity of comparing our work to DeepMapping, since it only performs on rigid scenes.
> > > Nevertheless, we will discuss DeepMapping in our paper since it deserves attention as being a novel optimization method for rigid registration.

---

> > > > ### Comment · Reviewer_rKbV · 2021-09-02
> > > > **Re: DeepMapping**
> > > >
> > > > Thanks for the response. To clarify: Taking into account the substantive comparisons against PointPWC-Net and FlowNet3D, I think a discussion of DeepMapping will be sufficient. I mentioned the numerical comparison to DeepMapping as it will be very interesting to see how the proposed approach performs against DeepMapping and I don't see a reason why it can't be compared (a non-rigid approach should still work on rigid scenes after all). However, not having this comparison is not a reason for rejecting the paper.

---

### Official Review · Reviewer_XKk4 · 2021-07-16

**Rating:** 7
**Confidence:** 4

**Summary:**

This paper studies the problem of scene flow: estimating 3D motion fields between given point clouds from a dynamic scene. The proposed method predicts scene flows by optimizing a neural network at execution time, using randomly initialized weights.

This approach of using a network architecture as a prior is shown to have better generalization performance for out-of-distribution samples compared to learning-based methods which require offline datasets. In contrast to another recently introduced non-learning approach which uses graph Laplacian priors, this method provides a continuous scene flow representation, and has better time and accuracy performances in large scale scenes.


**Ethical Concerns:**

There are no ethical issues with this paper, in my opinion.

**Limitations And Societal Impact:**

The authors have assessed the limitations of their work, including a failure case due to the nearest-neighbor distance term in the loss function. Figure 6 shows an example of this failure mode. Other than that, the paper contains an analysis for the degradation of the computation-time performance of the presented method, along with the baseline’s performance.

The paper also includes a broader impact section, for discussing potential outcomes of this work.


**Main Review:**

This is a well-written paper, which I enjoyed reading. While the problem studied in the paper has been around for a long time, the approach for solving the task is novel and interesting. The claims in the paper are supported by comprehensive experimental results across four different datasets.

My questions to the authors along with some comments are listed below.

- Selection of MLP networks to represent scene flows: Processing point clouds (or point sets) with MLPs has a few drawbacks that are already well-known, such as imposing an ordering of the points and requiring a fixed cardinality for the point set. While the distance function used in Eq. 4 allows point sets of different sizes, the MLP inputs point clouds of fixed size (set as 2048 in the experiments). A discussion about how to take point clouds of differing sizes to compute the scene flow would be useful -- especially for the case when computing over the sequence of M point sets.

- Related to the previous comment, could optimizing the network weights at execution time using an architecture designed to operate on point clouds (e.g. [1]) regularize the motion field better? Alternatively, can the method be modified to take a single point as input to the MLP at a time for allowing variable point cloud sizes?

- Can you comment on the computation/inference time of other learning- and non-learning-based methods? Adding how much time the non-rigid ICP and learning-based take would be helpful.

- What is meant by the rigid accumulation of point clouds, shown in Figure 7? Does it refer to computing the flow in a discrete fashion using nearest neighbors as opposed to the continuous flow estimation?


Minor comments:

- In line 110, I believe the reference to [15, 24] should be [15, 27] instead.
- You may consider adding a zoomed inset image in the top-down views of Figure 5 for better visual clarity.


References

1- Qi, C.R., Yi, L., Su, H. and Guibas, L.J., 2017. Pointnet++: Deep hierarchical feature learning on point sets in a metric space. arXiv preprint arXiv:1706.02413.


**Time Spent Reviewing:**

4

---

> ### Author Response · Authors · 2021-08-10
> **Response to Reviewer XKk4**
>
> We appreciate reviewer XKk4 for the thoughtful comments. We are happy to know our idea is novel and interesting and that the paper is well written and with comprehensive experiments.
>
> Here we address the comments.
>
> **Q1: On the selection of MLP networks to represent scene flows.**
>
> **A1:** Interesting observation. In the experiments in Table 1, the input pair of point clouds have the same number of points (randomly sampled during data loading). In the experiments of section 4.3 (estimating scene flow from large point clouds), the number of points in the input pair is arbitrarily different as we used the full raw lidar point clouds.
> Our method runs an optimization per scene, and the MLP regularizer is automatically instantiated at the beginning. Please keep in mind that a single point is an input to the MLP. Therefore, our method can naturally accommodate input pairs of point clouds with different sizes.
> If one would use our proposed objective function to train a self-supervised model, then having equal size point clouds would be more practical for batch learning. We are happy to make this clear in the paper.
>
> **Q2: Can different architecture be employed?**
>
> **A2:** Absolutely. What is the optimal architecture for scene flow regularization is still an open question. We found that minimal MLPs are enough to achieve compelling and competitive scene flow estimations. We have not explored an architecture similar to PointNet++ [1]. It would be interesting to see its capacity for scene flow regularization, but at the cost of more complicated/slow operations (*e.g*., nearest/farthest neighbor sampling and grouping).
>
> **Q3: On the computation/inference time of other methods.**
>
> **A3:**
> We acknowledge that inference time for supervised-based learning methods is faster than runtime optimization.
> The learning-based methods that optimize during training have very fast inference time (in the orders of milliseconds). While the non-learning-based methods that optimize during runtime have slow inference time (in the orders of seconds).
> Non-rigid ICP has a similar inference time to the Graph prior method. Please refer to Table 2 for more details.
> As we discussed before, our method is not yet competitive to learning-based methods *w.r.t* inference time as it relies on iterative optimization during runtime.
>
> **Q4: What is meant by the rigid accumulation of point clouds, shown in Figure 7?**
>
> **A4:** In short, rigid means that the point cloud accumulation was performed using a rigid registration method (*i.e*., ICP). Rigid 6-DoF poses are used for the registrations. We will clarify that in the paper.
>
> To give you a full picture, imagine an autonomous vehicle is moving through the world and sampling point clouds using a lidar sensor. Lidar point clouds are generally sparse, and often one wants to accumulate/densify a set to draw more geometrical information. Therefore we can register/align consecutive point clouds into a single frame. If the world is rigid, we can use the ICP algorithm for the pair-wise rigid registration (*i.e*., find the rigid 6-DoF pose that best aligns one point cloud against another). However, if the scene is dynamic and there are a lot of movers (*i.e*., moving pedestrians and/or vehicles), if we were to apply ICP we would see a "rolling shutter effect" (a smearing effect) on the dynamic objects. This effect is demonstrated in the front car in the middle row of Figure 7, where we used simple rigid 6-DoF poses to register the point clouds rigidly.
> On the other hand, scene flow gives a per-point transformation (*i.e*., a translational vector) that allows for non-rigid registration. Thus, reducing the smearing effect because the rigid and non-rigid parts of the scene are correctly registered/accumulated.
>
> > **Regarding minor comments.**
> 1. Yes, it should be [15,27] in L110.
> 2. We will consider adding a zoomed inset image in Figure 5 for clarity.
>
> **References:**
> 1. Qi, Charles R., Li Yi, Hao Su, and Leonidas J. Guibas. "Pointnet++: Deep hierarchical feature learning on point sets in a metric space." arXiv preprint arXiv:1706.02413 (2017).

---

> > ### Comment · Reviewer_XKk4 · 2021-08-30
> > **Response to the authors**
> >
> > Thank you for the response. The reviewers’ concerns have been mostly addressed, therefore I vote for the acceptance of this paper. I encourage the authors to add the experiments from the rebuttal to the final version to further strengthen the paper.

---

### Official Review · Reviewer_8oZd · 2021-07-18

**Rating:** 7
**Confidence:** 4

**Summary:**

This submission presents a method for estimating scene flow between two point clouds that uses a neural network as an implicit regularizer during optimization. The submission revisits the classical optimization-based approach where instead of training a neural net to estimate flow for a given scene, it optimizes the flow per scene and does not need to train the network beforehand. Instead, the network is “trained” to produce flow for the given scene, such that representing the flow with the network introduces some form of implicit smoothness regularization.

**Limitations And Societal Impact:**

Limitations are not fully discussed (see above).

**Main Review:**

# Originality

The idea presented in the submission is interesting and while neural networks have been applied as implicit functions in other domains, their application to scene flow is novel as far as I know. The presented method connects modern neural network based approaches to flow estimation to classical flow estimation approaches. The listed related work is extensive. However, I believe that the comparison to related approaches has important gaps – I will give more details about this in the next point.

# Quality

The description of the method is technically sound. The experimental evaluation of the proposed method and its components (apart from the comparison to other work in Table 1) is extensive and insightful, e.g. Sections 4.1, 4.3, 4.4, and 4.6. I also appreciate the discussion of the failure case in Section 4.5 – more examples would be even better (maybe in the supplemental material). I find Figure 5 especially helpful to understand how the model is optimized over time and how it interpolates between points. From that figure, I’m curious if the removal of the ground plane is essential for the method to work well. Maybe the authors could comment on that if they have done those experiments.

Unfortunately, some claims seem overly general and are not well supported by evidence: a) L26-28 portrays the need for data in itself as a problem, including training on synthetic scenarios and then fine-tuning on small real datasets. But this in itself is not a problem, at least not for domains where such data exists, as for autonomous driving which this paper focuses on. b) L29 claims that poor generalization is a problem but fails to give evidence for this in general. Yes, generalization is poor when one goes from FlyingThings3D to KITTI, but this looks like a strawman because the more relevant question would be how well the models generalize between KITTI, nuScenes, Argoverse, and other autonomous driving scenarios. In contrast to the statement here, e.g. [28] states that “we show that our network, even trained on synthetic data, is able to robustly estimate scene flow in point clouds from real scans, showing its great generalizability.” c) L13-15 make the argument that because optimization happens at runtime and no offline data is needed, this approach would be ideal for deployment in autonomous driving – ignoring the problem that optimization at runtime is much slower compared to performing optimization during training time, which will in fact make it very challenging to apply this method for autonomous driving to produce scene flow in real time. Based on Figure 3, inference takes > 5s for a single scene. Here the discussion should be more balanced and mention both the advantages and the disadvantages of performing optimization at runtime. Figure 1 is also misleading in this regard as it cherry-picks one alternative method that is slower but disregards other related methods that are much faster than the proposed approach. L168 states that the evaluation focuses on both accuracy and computational cost but does not take the cost at inference into account when comparing to other methods in Table 1.

The comparison to SOTA in Table 1 does not appear fair to me for a number of reasons:
a) The SOTA comparison only includes a single (and no longer SOTA) supervised method FlowNet3D and ignores important related work with published results on FlyingThings3D and KITTI, e.g. HPLFlowNet (CVPR 2019), PointPWC-Net (ECCV 2020), FLOT (ECCV 2020)
b) The supervised method FlowNet3D is only evaluated with domain transfer from FlyingThings3D to driving scenes even though a model trained on KITTI is available. This is stated in the text but not in the caption of Table 1 making it easy to miss. Testing generalization is important but to give the full picture, the KITTI-trained model should be included and it should be made clear when a model is tested outside of the training domain.
c) Self-supervised methods here are only optimized on the training scenes and need to generalize to the test scenes. For a fair comparison, it would be good to also report their scores after fine-tuning on the test scenes in order to have access to the same information that the presented method has during optimization.
d) To adequately show the advantages and disadvantages of these methods, an estimate of the inference time should be added to the table as this will vary strongly between the compared methods and orders of magnitude differences in inference time substantially change the applicability of a method for estimating motion (which often needs to be done in real time).

L162: I’m not convinced that comparing the number of parameters is very useful here. Parameters have very different semantics in both cases. For a learned model, they correspond to the memory required to store the model, but in the proposed method, they need to be optimized for inference. As the learned model represents a flow estimation model for “all” scenes it should not be surprising that this needs to be larger than a model that represents the flow for a single scene. If one wants to make the comparison, both memory and inference time should be taken into account.

# Clarity

The submission is written clearly and well organized. All code is included in the supplementary material, which should make it easy to reproduce the work.

# Significance

The presented method and results are important to the community and could be the basis for further exploration into this approach of estimating scene flow during inference using neural networks as implicit functions. My main reservation regarding this work regards the comparison to existing learning-based approaches both in the text and in the empirical results. Here the paper seems to focus too much on the advantages of the presented method and does not always give the whole picture.

To make this very clear: I think that the paper has very valuable contributions as it shows progress along a direction that is not the main path taken by related work right now. Even if a complete comparison to related work (as described above) shows that this approach is not the most practical method (in terms of accuracy and / or inference time) it would still be a very valuable contribution because it shows progress along an approach that is very different from most existing work. But only with a fair portrayal and comparison of related work will I be able to recommend accepting the paper.

EDIT: My main concerns have been addressed in the post-review discussion with the authors.

**Time Spent Reviewing:**

3.5

---

> ### Author Response · Authors · 2021-08-10
> **Response to Reviewer 8oZd**
>
> We appreciate reviewer 8oZd for the thoughtful comments. We are encouraged to know our idea is novel, interesting, and has very valuable contributions, and that the paper is well written and organized.
>
> We address the main concerns below.
>
> > **On the removal of ground plane points.**
>
> This is an important point to be discussed. The majority of the methods (*e.g*., FlowNet3D) for scene flow prediction from point clouds sampled from lidar sensors (*e.g*., KITTI Scene Flow dataset) have found that the ground points negatively impact the performance. The ground is a large piece of flat geometry with little cue to predict motion. Imagine trying to find correspondences in a large flat white wall to compute optical flow. It is intractable without a very large context. We observe the same aperture problem during scene flow estimation. Also, lidar point clouds from driving scenes have a specific ground sampling pattern that resembles an arch of points every other meter. If not removed during scene flow prediction, those points will be snapped/stitched to the closest arch of points and thus biasing to much the end-point-error metric. We believe this problem should be further investigated and ideally handled by the method without prior semantic knowledge of "ground."
>
> > **On confusions about our claims.**
>
> **a)** We respectfully disagree with the reviewer claiming that scene flow data exists for the autonomous driving domain. We agree that there exist real-world, large-scale point cloud datasets sampled from lidar sensors (*e.g*., KITTI, nuScenes, Argoverse, Waymo Open). However, there still do not exist ground-truth scene flow labels for such large-scale datasets. There are pseudo-ground-truth labels for a fraction of these datasets collected using the methods described in the FlowNet3D and Graph prior papers. That is why we believe there is a growing interest in self-supervised methods to unlock scene flow representations from those large-scale point cloud datasets themselves without any annotations. Our method is posed as a runtime scene flow optimization to handle any non-annotated point clouds from any domain.
>
> **b)** We believe that poor generalization to out-of-the-distribution data is a widespread problem in deep learning.
> We quantified this problem for the scene flow task in Table 1. If we train a supervised scene flow model such as FlowNet3D using the large-scale synthetic FlyingThings dataset, then test the model on the same domain, the results are impressive -- given that the FlyingThings is a very challenging dataset (values shown in bold in the top left in Table 1).
> Suppose we test the same model (*i.e*., trained only on FlyingThings) on other domains that are statistically different, for example, the KITTI Scene Flow dataset. In that case, the results are not as impressive and might be catastrophically wrong. One might argue that an EPE value around 0.19 with low accuracy is a fair generalization. However, our non-learning-based method is orders of magnitude better in all metrics.
> Please also note that our experiments were similar to what was proposed in the FlowNet3D paper, where the authors claimed the method generalizes to other data domains.
>
> **c)** We acknowledge that the runtime optimization is much slower when compared to the inference time given by supervised methods. We have shown that our proposed scene flow objective function can be directly optimized during runtime to achieve robust results and generalize to other domains.
> Nevertheless, our method could also be explored to self-supervise the learning of scene flow from point clouds using, for example, an off-the-shelf scene flow model. The tradeoff will depend on the application.
> For example, suppose robustness/generalizability is not an issue but rather the inference time. In that case, our proposed objective function can be used to train a self-supervised model and act as a surrogate of our non-learning method but inheriting the faster inference from a trained model.
> Another example is that our method can be applied to robotics applications like offline mapping and point cloud accumulations, *etc*.
> We will further discuss such tradeoffs between the accuracy and computational time in Section 4.3.
>
> We respectfully disagree with the comment about Figure 1. We focused on comparing two optimization methods (graph prior and our neural scene flow prior) without any learning involved. The learning-based methods are optimized during training while ours are optimized during run/test time which is definitely much slower.
>
> > **On the comparisons to SOTA.**
>
> **a)** We acknowledged these works and have mentioned them in the related work section (we will reference HPLFlowNet). We compared PointPWC-Net in the self-supervised setting in Table 1. We found that HPLFlowNet and FLOT have reported similar to worse results than PointPWC-Net on FlyingThings3D and KITTI (refer to the original PointPWC-Net paper where they compared against HPLFlowNet, and to the FLOT paper where they compared against HPLFlowNet and PointPWC-Net). Therefore, we only chose PointPWC-Net for our experiments. Also, please note that our method is non-learning-based, and we think it is fair to choose a single baseline as FlowNet3D that is simple and widely used in the community.
>
> **b)** We agree. We will add text to the caption of Table 1 to clarify it.
>
> **c)** Yes, we had already done what the reviewer suggested. Please refer to L194-L202 for the detailed training setups of the supervised and self-supervised methods.
>
> **d)** As previously discussed, we will further elaborate on the tradeoffs between accuracy/robustness and inference time for our method.
> We would like to clarify that we are not claiming our proposed method is suitable for real-time applications in its current research phase. However, it will be an exciting direction to investigate in the future.
> Instead, our focus is the runtime optimization, which differs from other supervise-based learning methods where optimization only occurred at training time. The motivation of our work is to find a dataless method that does not rely on any training data nor labels for estimating robust scene flow from point-cloud pairs.
> Also, it is important to notice that many applications do not rely on real-time scene flow estimation. We gave an example of lidar point cloud densification/accumulation to generate denser point clouds that can be used for mapping, creating denser depth maps, etc. For example, the recently released Argoverse Stereo dataset (https://www.argoverse.org/data.html#stereo-link) has been built using the Graph prior method. Consecutive lidar point clouds were accumulated using runtime scene flow optimization and then projected onto stereo imagery to generate lidar-based ground-truth depth labels.
>
> > **On the number of parameters not being useful (L162).**
>
> We reported the number of parameters to show the simplicity of our proposed method when compared to FlowNet3D.
> Even with a simple MLP architecture, our method still achieves impressive results on several datasets without any training data supervision.
> But we agree with the reviewer that we should have taken into account the inference time for a fair comparison.
> We will further discuss the tradeoffs in the paper.
>
> In the end, we thank you for pointing out that the limitations were not fully discussed. We will add full discussions in the main paper to address your concerns.

---

> > ### Comment · Reviewer_8oZd · 2021-08-16
> > **Response to Response**
> >
> > Thank you for your thorough reply. I appreciate the clarifications. I will comment on a subset of the aspects discussed above.
> >
> > **Claims a-b)** I fully agree that there is a lot of value looking into approaches that need less or no annotation. My main concern is about providing a balanced picture. Including the supervised method trained on KITTI in addition to the one trained on FlyingThings would be much more informative as it shows generalization to nuScenes and argoverse when the domain shift is more and less severe. Since there exists some annotated data in the autonomous driving domain, we cannot accurately portray the supervised approach without showing how it performs using such data.
> >
> > **Claims c)** I am not saying that there is no space for this method to be very useful in practice, especially when used in an offline fashion. I was only expressing my concerns about that the much larger runtime is not included in the tables and not discussed in comparisons to SOTA. Yes, Figure 1 is only comparing two optimization-based methods but it is placed in the beginning of the paper and referenced in L65-67 right after stating that results are competitive with supervised, self-supervised, and non learning methods. If used in this spot, I think Figure 1 should include runtime estimates of those other methods to give the full picture. The very least that should be done is to clarify both in the L67 and in the caption that this Figure only compares to methods that perform optimization at inference and that other methods are (10x - 100x ?) faster. Again, I'm not saying that the method is not useful because it is slow. I think the method is very valuable. But when relating it to other approaches, this needs to be clearly communicated to the reader including in the intro and in the head figure.
> >
> > **SOTA a)** But PointPWC-Net reports numbers for supervised and self-supervised training. It is not okay to misrepresent supervised SOTA by only reporting numbers of a worse model -- FlowNet3D is worse than HPLFlowNet and PointPWC-Net on Flyingthings3D (EPE of 0.1136 vs 0.0804 vs 0.0588)? Upon checking those numbers, I also found that the PointPWC-Net's numbers are actually better than reported in the submission, e.g. EPE of 0.043 m for KITTI instead of the reported 0.177. How can that be the case? The PointPWC-Net paper also reports numbers for FlyingThings3D, which are omitted in this submission, that are better than the proposed method. What am I missing here?
> >
> > **SOTA c)** L194-202 Does not explain what I was suggesting. I was suggesting to add a variant that finetunes on the test scenes to be more comparable with the proposed method. The text describes training on the same domain, but not on the test split.
> >
> > **SOTA d)** I can very much relate to this motivation and I hope that the paper will be able to better represent and discuss these trade-offs.

---

> > > ### Author Response · Authors · 2021-08-17
> > > **Response to additional comments by reviewer 8oZd**
> > >
> > > We thank you for the additional comments.
> > > We hope the comments below bring further clarifications.
> > >
> > > **Claims a-b)**
> > > We agree that providing a balanced picture is important.
> > > We want to reclaim that such an experimental setting is motivated by that readily available annotated point clouds do not always exist to fine-tune supervised scene flow models to a specific domain. Such an experiment setting to test generalizability on the out-of-the-distribution real-world dataset is well-known and widely used. We want to refer the reviewer to the FlowNet3D[1], PointPWC-Net[2], FLOT[3], HPLFLowNet[4], Graph prior[5] paper. These papers perform/compare supervised methods that are trained on FlyingThings3D and directly tested on the KITTI dataset to test the generalizability (we provide detailed reference below).
> > > Please note, even for FlowNet3D, the authors only report finetuning on a small (100 scenes) KITTI dataset for ground point ablation study (FlowNet3D Table 5), not a thorough experiments for comparison to other methods.
> > > Another fact is that even for the KITTI dataset, there only exist 200 annotations for scene flow estimation (as claimed in FlowNet3D, Section 6.1, 1st paragraph). These learning-based methods are data-driven, and cannot rely on these small annotations to directly train a large real-world dataset.
> > > Current learning-based methods, as far as we know, need to train at least on large synthetic datasets to provide adequate supervision, then directly test on large real-world datasets or fine-tune using self-supervision.
> > > As we claimed before, we are confident that such an experimental setting is reasonable, widely used, and is able to test the out-of-the-distribution generalizability that is important in the scene flow estimation when large-scale annotations are not available.
> > >
> > > > **Detailed reference to those papers mentioned above.**
> > > 1. FlowNet3D: Section 6 Experiments, 1st paragraph; Section 6.2  Generalization to Real Lidar Scans in KITTI; Table 4.
> > > 2. PointPWC-Net: Section 5 Experiments, 1st paragraph, the authors only fine-tuned their self-supervised method on KITTI, since the annotations for KITTI are small (only 200 as stated before), they only fine-tuned using self-supervision; Table 1, only self-supervised PointPWC-Net is fine-tuned using self-supervision on KITTI.
> > > 3. FLOT: Section 4.1 Datasets under experiment section; Table 1.
> > > 4. HPFLowNet: Section 5 Experiments, 1st paragraph 2); Section 5.2,  Generalization results on real-world data; Table 1.
> > > 5. Graph prior: Section 5 Experiments Implementation details; Table 1 and its caption.
> > >
> > >
> > > **Claims c)**
> > > Thank you for the suggestion. We will specify more clearly that Figure 1 is the comparison of non-learning-based methods.
> > > We are also happy to improve our claims by adding discussions about inference time tradeoffs as we commented before.
> > > We would like to further clarify that in L67 when we mentioned Figure 1, we claimed: "our method scales to large real-world point clouds while achieving great accuracy and time complexity (Fig. 1)". One of the advantages of our method is that it scales to large point clouds with high density.
> > > As we claimed before, training supervised/self-supervised models with high-density point clouds is not always practical due to high memory consumption. Typically, such models are trained with up to ~8k points.
> > > In HPLFlowNet[4], the authors found that their method train on less number of points is able to generalize to denser point clouds, but does not improve a lot (HPLFlowNet, Table 3).
> > > However, our method can gain a great improvement in performance with much denser point clouds (see Figure 3 and Table 2).
> > > We will add these discussions to the paper and avoid misunderstandings in our claims in L67.
> > >
> > >
> > > **SOTA a)**
> > > We acknowledge that in the original PointPWC-Net paper, the authors compared different settings in Table 1, -- full supervision, self-supervision, self-supervision + self-supervision, and full supervision + self-supervision. However, we only chose one setting (*i.e*., full supervision + self-supervision) when comparing our method to PointPWC-Net. In PointPWC-Net Table 1, the full supervision + self-supervision is the best setting for the KITTI dataset. We followed this setting to test on nuScenes and Argoverse Scene Flow datasets for PointPWC-Net and Just Go with the Flow. Since for experiments on FlyingThings3D, there involves no self-supervision, we did not report results for self-supervised methods under this setting.
> > >
> > > Please also note that in the original PointPWC-Net paper, the authors used ***8,192*** points for the input point cloud (in Section 5 Experiment Implementation Details, Page 11), while in our experiment, we used ***2,048*** points for Table 1 (in L230 and Table 1 caption). It is natural that there exists a performance gap between our experiment and theirs. Again, these results are not first reported by us but have already been reported in [5].
> > > We use ***2,048*** that followed experiment settings in FlowNet3D[1] and Graph prior[5] paper to constrain a rather small model for comparison. In contrast, our model has fewer parameters and consumes fewer memories.
> > > Note that our method, although simple and tested on sparse point clouds (***2,048*** points), achieved impressive results on various datasets.
> > >
> > > Here we provide the supervised PointPWC-Net results below. We will add these results to Table 1.
> > >
> > >
> > > #### Comparison with supervised PointPWC-Net (including FlowNet3D results on FlyingThings3D for reference)
> > >
> > > >FlyingThings3D
> > >
> > > |       | $\mathcal{E}(m)\downarrow$ | $Acc_{5}($%$)\uparrow$ | $Acc_{10}($%$)\uparrow$ | $\theta_{\epsilon} (rad)\downarrow$ |
> > > | :---: | :----: | :---: | :----: | :---: |
> > > | FlowNet3D | 0.134 | 22.64 | 54.17 | 0.305 |
> > > | PointPWC-Net | **0.121** | **29.09** | **61.70** | **0.229** |
> > > | Ours   | 0.234 | 19.16 $\pm$ 0.23 | 46.74 $\pm$ 0.46 | 0.341 |
> > >
> > >
> > > >nuScenes
> > >
> > > |       | $\mathcal{E}(m)\downarrow$ | $Acc_{5}($%$)\uparrow$ | $Acc_{10}($%$)\uparrow$ | $\theta_{\epsilon} (rad)\downarrow$ |
> > > | :---: | :----: | :---: | :----: | :---: |
> > > | PointPWC-Net | 0.442 | 7.64 | 22.32 | 0.497 |
> > > | Ours   | **0.175 $\pm$ 0.01** | **35.18 $\pm$ 1.32** | **63.45 $\pm$ 0.46** | **0.279 $\pm$ 0.04** |
> > >
> > >
> > > >KITTI
> > >
> > > |       | $\mathcal{E}(m)\downarrow$ | $Acc_{5}($%$)\uparrow$ | $Acc_{10}($%$)\uparrow$ | $\theta_{\epsilon} (rad)\downarrow$ |
> > > | :---: | :----: | :---: | :----: | :---: |
> > > | PointPWC-Net | 0.142 | 29.91 | 59.83 | 0.239 |
> > > | Ours   | **0.050 $\pm$ 0.01** | **81.68 $\pm$ 2.00** | **93.19 $\pm$ 1.30** | **0.133 $\pm$ 0.01** |
> > >
> > >
> > > >Argoverse
> > >
> > > |       | $\mathcal{E}(m)\downarrow$ | $Acc_{5}($%$)\uparrow$ | $Acc_{10}($%$)\uparrow$ | $\theta_{\epsilon} (rad)\downarrow$ |
> > > | :---: | :----: | :---: | :----: | :---: |
> > > | PointPWC-Net | 0.405 | 8.25 | 25.47 | 0.674 |
> > > | Ours   | **0.159 $\pm$ 0.01** | **38.43 $\pm$ 0.48** | **63.08 $\pm$ 0.59** | **0.374 $\pm$ 0.01** |
> > >
> > >
> > > **SOTA c)**
> > > We kindly remind the reviewer to take a closer look at L196-202. We clearly stated that "**Just Go with the Flow** and **PointPWC-Net:** trained on FlyingThings3D with supervision and fine-tuned on domain-matched datasets with self-supervision (*ie*., fine-tuned and tested on statistically similar data)." That is saying, for self-supervised methods, we first trained on FlyingThings3D, then fine-tuned and tested on KITTI, nuScenes, Argoverse respectively. We will make this sentence clearer and more specific in the paper.
> > >
> > > **SOTA d)**
> > > Thank you for the comment, we will elaborate on these trade-offs.
> > >
> > >
> > > **References:**
> > > 1. Liu, Xingyu, Charles R. Qi, and Leonidas J. Guibas. "Flownet3d: Learning scene flow in 3d point clouds." Proceedings of the IEEE/CVF Conference on Computer Vision and Pattern Recognition. 2019.
> > > 2. Wu, Wenxuan, Zhiyuan Wang, Zhuwen Li, Wei Liu, and Li Fuxin. "Pointpwc-net: A coarse-to-fine network for supervised and self-supervised scene flow estimation on 3d point clouds." arXiv preprint arXiv:1911.12408 (2019).
> > > 3. Puy, Gilles, Alexandre Boulch, and Renaud Marlet. "Flot: Scene flow on point clouds guided by optimal transport." Computer Vision–ECCV 2020: 16th European Conference, Glasgow, UK, August 23–28, 2020, Proceedings, Part XXVIII 16. Springer International Publishing, 2020.
> > > 4. Gu, Xiuye, Yijie Wang, Chongruo Wu, Yong Jae Lee, and Panqu Wang. "Hplflownet: Hierarchical permutohedral lattice flownet for scene flow estimation on large-scale point clouds." Proceedings of the IEEE/CVF Conference on Computer Vision and Pattern Recognition. 2019.
> > > 5. Pontes, Jhony Kaesemodel, James Hays, and Simon Lucey. "Scene Flow from Point Clouds with or without Learning." 2020 International Conference on 3D Vision (3DV). IEEE, 2020.

---

> > > > ### Comment · Reviewer_8oZd · 2021-08-20
> > > > **Response to response**
> > > >
> > > > Thank you for the additional information and clarification. To not drag this out too long, I'll just give a brief reply here.
> > > >
> > > > SOTA a) If the current best methods on this problem use the full point set of 8,192 points, I see a problem with changing the benchmark definition to only use a subset of these points -- yes, this has been done in prior papers, but that does not make it right. At the very least, this needs to be made transparent to the reader. The problem that I have with this is that I cannot be sure that the performance gap to the results published on 8,192 point input is actually only due to the changing number of points or if the method was not correctly reproduced and if the latter is true, I cannot trust the comparison on the other datasets either. Since PointPWC-Net is applied to novel data here, showing that the used implementation reproduces the published results is essential for trusting that implementation.
> > > >
> > > > SOTA c) This is a good point and it makes me realize that these methods should not be referred to as "self-supervised" in the table, but rather "semi-supervised" because they use both supervised and self-supervised learning. The thing you are arguing against was not the point I wanted to make here though. I was suggesting that there should be an additional comparison in that table to these methods when they are fine-tuned on the *exact test data* (not the training data from the same distribution). This would be a fairer comparison to the presented method as optimization can also be done during inference here. I would be good to try out both optimizing the baselines for all test scenes as well as optimizing the method for individual scenes.

---

> > > > > ### Author Response · Authors · 2021-08-25
> > > > > **Response to additional comments by reviewer 8oZd**
> > > > >
> > > > > Thank you for the additional comments.
> > > > > We hope the comments below bring further clarifications to your concerns.
> > > > >
> > > > > **SOTA a)**
> > > > > We have specified that we used ***2,048*** points both in the text L230 and Table 1 caption. Upon your suggestion, we will modify L230 to "We set the number of points to 2,048, which follows the experimental protocols as in FlowNet3D [28] and Graph prior [43]", to make it clearer to the reader.
> > > > >
> > > > > We would like to further clarify that we used the original PointPWC-Net implementation published by the authors but we used the two pre-processed datasets provided by FlowNet3D: FlyingThings3D and KITTI.
> > > > > In the PointPWC-Net’s data loader implementation (https://github.com/DylanWusee/PointPWC/blob/master/transforms/transforms.py#L137 and https://github.com/DylanWusee/PointPWC/blob/master/config_evaluate.yaml#L28) there is a threshold to crop a region of the lidar point cloud that is within 35 meters of depth along the y-axis.
> > > > > We did not use such a threshold, we used all points available that range up to around 85 meters away from the center of the scene where the autonomous vehicle was located. Lidar point clouds get sparser as distance increases, so it is more difficult to estimate scene flow in far away, sparser regions.
> > > > > Nevertheless, we did not shy away from this fact in our experiments. Unfortunately, the scene flow community does not yet have well-established benchmarks to serve as common ground. We really hope to contribute to this space in the future.
> > > > >
> > > > > We performed a new experiment with the hope that it would help clarify the reviewer's concerns on the point density.
> > > > > We first tried to use a pretrained PointPWC-Net model (pretrained on FlyingThings3D) released by the authors (note that we used the same model implementation for the experiments in our paper) to directly test on the KITTI dataset we presented in the paper.
> > > > > However, this model is pretrained on a dataset that only has points within 35 meters of depth, when tested on our full-range point cloud data, it completely failed.
> > > > > To test the influence of the point density, we then tested PointPWC-Net using this pretrained model (available here: https://github.com/DylanWusee/PointPWC/tree/master/pretrain_weights) and their preprocessed KITTI data with points only within 35 meters of depth.
> > > > > To make a fair comparison, we also compared the results with our method on the same dataset.
> > > > > We show the results below.
> > > > > Note that the pretrained model provided by the PointPWC-Net's authors was fine-tuned after the paper submission as mentioned in the official PointPWC-Net GitHub repository (https://github.com/DylanWusee/PointPWC#pointpwc-net-cost-volume-on-point-clouds-for-self--supervised-scene-flow-estimation).
> > > > >
> > > > > >KITTI (200 scenes, with points only within 35 meters of depth)
> > > > >
> > > > > |       | $\mathcal{E}(m)\downarrow$ | $Acc_{5}($%$)\uparrow$ | $Acc_{10}($%$)\uparrow$ | $\theta_{\epsilon} (rad)\downarrow$ |
> > > > > | :---: | :----: | :---: | :----: | :---: |
> > > > > | PointPWC-Net (reported in their paper, ***8,192*** points) | 0.0694 | 72.81 | 88.84 | - |
> > > > > | PointPWC-Net (their pretrained model, ***8,192*** points) | 0.0778 | 82.24 | 90.96 | 0.1127 |
> > > > > | Ours (***8,192*** points)  | **0.0493** | **90.38** | **95.27** | **0.1099** |
> > > > > | PointPWC-Net (their pretrained model, ***2,048*** points) | 0.1298 | 58.14 | 79.85 | 0.1562 |
> > > > > | Ours (***2,048*** points)  | **0.0504** | **85.27** | **94.66** | **0.1209** |
> > > > >
> > > > >
> > > > > These results reveal three facts: 1. the point density does greatly affect the performance of PointPWC-Net; 2. our method, although simple and tested on sparse point clouds, achieves better accuracy; 3. our dataset (following original FlowNet3D, Graph Prior, and other papers) contains large-range raw point clouds that are more challenging than the dataset used in the PointPWC-Net paper.
> > > > >
> > > > >
> > > > > **SOTA c)**
> > > > > Thank you for further clarifications on the question.
> > > > > Yes, those experiments would be interesting. However, we think setting a simple coordinate-based network of a simple MLP as we did, already showed the potential of using neural networks as priors for the scene flow estimation task. Complex priors, based on FlowNet3D, PointPWC-Net architectures might perform well (see Graph Prior paper, Table 1, where the authors used PointPWC-Net to do non-learning-based optimization) but at the cost of large memory consumption with too many layers of convolutions, sampling, and grouping operations. It would also be more difficult to optimize during runtime with a large number of parameters.

---

> > > > > > ### Comment · Reviewer_8oZd · 2021-08-25
> > > > > > **Response to Response**
> > > > > >
> > > > > > Thank you for running these additional experiments and digging into the details of PointPWC-Net. This addresses my main concern and especially the information about the depth threshold is central to make sense of the different results.
> > > > > >
> > > > > > Although this back and forth was a bit of a struggle, I feel that the paper will be *substantially stronger* with these information included (and the other points discussed earlier). Assuming that the paper will be updated accordingly, I'm now happy to revise my rating and recommend accepting the paper.

---

### Official Review · Reviewer_y7UW · 2021-07-20

**Rating:** 6
**Confidence:** 3

**Summary:**

This paper studies scene-flow estimation. Following the idea of deep-image prior, it proposes an unsupervised way to learn scene flow. The results are competitive, or even better than supervised flow-estimation methods trained on very different training set. Moreover, it also out-performs existing optimization based unsupervised flow-estimation methods.

**Ethical Concerns:**

I have read the Ethics Guidelines, and I don't find any concerns from this paper.
1. Potential negative societal impacts: It doesn't involve any social experiments, and doesn't use human-derived data.
2. General ethical conduct: the dataset used in this work are public ones.

**Limitations And Societal Impact:**

1. Limitations: one failure case is shown in Sec 4.5 and Fig.6 to show the limitation of the NN loss.
2. Societal impacts are properly discussed (L309ff).

**Main Review:**

I'm not an expert in the field of 3D flow estimation, and not super familiar with the SOTA unsupervised or self-supervised methods for this tasks. It's likely that my interpretation or understanding is biased or wrong. For the paper authors and other people, feel free to correct me if you find such things.

Strength:
1. This paper identifies the poor generalizability of one established supervised method. The results are supervised and it performs very poor on new data domain. If one or two more baselines could be studied, this result would be more convincing.
2. The proposed method out-performs existing work "graph prior" significantly and consistently. Another good property of this work is that many prior-deep learning works are mentioned or compared, which is rarely now days and show be encouraged.
3. The experiments are conducted on multiple different experiments, and promising results are presented and visualized.
4. Writing is mostly clearly and easy to follow.

Weakness:
1. The supervised baselines seem to be very weak. On the unseen dataset, the Acc_5 could be as low as 1.34%. Even on the training set (FyingThings), Acc_5 is only 22.64%. As a comparison, the proposed method can achieve 80% Acc on other dataset. Therefore, I don't think these supervised baselines are strong, which makes the results less convincing.
2. For 3D flow estimation, shouldn't EPE the major metric?
3. Recently, the RAFT model become the most popular method for 3D flow estimation. For 3D flow, RAFT-3D [1] is the current SOTA method. I think it's a more proper baseline than FlowNet3D, and I'm also wondering how well will it generalize to new dataset. In fact, the provides numbers/models for both FlyingThings and Kitti, which help us to understand both the generalizability and performance upper bound.
[1] RAFT-3D: Scene Flow using Rigid-Motion Embeddings. Teed and Deng. CVPR 2021
4. In Deep image prior, it tricky to find an optimal stopping iteration during training time. It's unclear to me how does this work handle this problem.
5. This work should be classified as self-supervised learning approach IMO. It's essentially corresponding learning, during which the 3d flow is also learned as an architectural prior.
6. The cycle-consistency loss is an important innovation but not ablated.
7. The architectures are inspired by recent implicit models but lack of motivation. I'm not convinced by the advantages of doing this and why this framework is proper for flow-estimation.

Misc / Questions:
1. In Fig.3, why does one line of graph priors decrease over time?
2. In Tab.1, mean/var of "graph prior" and "ours"should be reported for all metrics for completeness.
3. Deep image prior trains one model for each picture. Does this work also train one model for each scene?


**Time Spent Reviewing:**

7

---

> ### Author Response · Authors · 2021-08-10
> **Response to Reviewer y7UW**
>
> We appreciate reviewer y7UW for the thoughtful comments. We are happy to know our results are promising and the paper is well written and easy to follow.
>
> Here we address the main concerns.
>
> 1. We chose FlowNet3D as a baseline for our supervised experiments because of its simplicity and capacity to predict scene flow directly from point clouds. One of our motivations is to show that supervised scene flow methods do not generalize to statistically different domains. If we train FlowNet3D on the synthetic FlyingThings dataset (which contains randomly moving objects such as chairs, lamps, *etc*.) and test on real-world lidar point clouds from driving scenarios, the performance is poor. We would expect this also to be true for other supervised scene flow methods. These results are also reported in [2] showing the bad generalizability of supervised methods and the impressive performance of non-learning-based methods on out-of-the-distribution datasets. Such an experimental setting is motivated by the lack of large-scale scene flow annotations on real-world point clouds. Current scene flow methods train on unrealistic synthetic datasets to fine-tune on small-scale real datasets annotated with pseudo-ground-truth scene flow as described in [1,2]. Therefore, we argue that readily available annotated point clouds will not always exist to fine-tune supervised scene flow models to a specific domain. There are many different sensor types (*e.g*., lidars, depth sensors) with different sampling, range, and density patterns, which makes it difficult for supervised methods to generalize over all such different data. Also, please remember that we are not advocating a competing method to supervised or self-supervised methods. We propose a non-learning-based approach that draws inspiration from modern machine learning to optimize scene flow during test time that generalizes and scales better. Nevertheless, our proposed scene flow objective can also be explored as a self-supervisory signal to train off-the-shelf supervised scene flow models when ground-truth labels are unavailable.
>
> 2. Yes, the end-point error (EPE) is the principal metric used in scene flow research. We reported EPE and accuracy metrics derived from it. We described the metrics and their notation in detail in Metrics (L180) in the Experiments section.
>
> 3. We thank the reviewer for pointing out the supervised RAFT-3D method. However, RAFT-3D estimates scene flow from a pair of stereo or RGB-D images. Our method estimates scene flow directly from point clouds without any image information. Nevertheless, RAFT-3D deserves to be referenced in our paper.
>
> 4. Similar to the Deep Image Prior work, we also found that a simple early stopping strategy is enough to return good scene flow solutions. We tracked the optimization loss, and if no improvement is observed after some iterations (*e.g*., 100 iterations), the optimization is stopped, and the optimal parameters are returned. No improvement is defined by an absolute change in the loss of less than a threshold (*e.g*., 1e-5). We described the use of early stopping in the Implementation details (L189-L191) in the Experiments section.
>
> 5. We would like to clarify that our proposed method optimizes, at runtime, a scene flow objective function that is regularized by a neural network. Also, our proposed method optimizes the scene flow for a specific scene -- thus not relying on training data. Therefore, we would not classify it as a self-supervised method but as a ***non-learning-based method***.
>
> 6. We would like to clarify that the cycle-consistency loss is not an innovation in our work. Cycle consistency has been widely used in optical and scene flow research. Recent learning-based scene flow methods (*e.g*., FlowNet3D, Just Go with the Flow) have employed a cycle consistency term to further regularize the scene flow and achieved better results. We are happy to include in our supplementary material a simple ablation study to show the importance of the cycle consistency term. Please also refer to Section A.2 Backward flow in our supplementary material for details on the backward flow and a qualitative example showing the importance of cycle consistency.
>
> #### Comparison w/ or w/o backward flow
>
> >KITTI Scene Flow Dataset (2048 points)
>
> |       | $\mathcal{E}(m)\downarrow$ | $Acc_{5}($\%$)\uparrow$ | $Acc_{10}($\%$)\uparrow$ | $\theta_{\epsilon} (rad)\downarrow$ |
> | :---: | :----: | :---: | :----: | :---: |
> | Ours (w/o backward flow) | 0.113 | 60.36 | 78.85 | 0.191 |
> | Ours (w/ backward flow)  | **0.052** | **80.70** | **92.09** | **0.133** |
>
> 7. Our motivation is to propose a method to robustly estimate scene flow from a single pair of point clouds without any training data nor labels. We found that a simple neural network architecture is good enough to implicitly regularize the scene flow and provide smoothness to the predictions. Moreover, we would like to point out that our method can also be explored as a self-supervisory signal to train, for example, an off-the-shelf scene flow model in a self-supervised way. It would inherit the fast inference time of a trained model. Another essential property of our method is that it scales to denser point clouds. As shown in Experiment section 4.3, our method can estimate scene flow from point clouds with about 80k points (lidar point clouds from the Argoverse dataset) with high fidelity. In contrast, training supervised/self-supervised models with high-density point clouds is not always practical due to high memory usage. Typically, such models are trained with up to ~8k points.
>
> Here we would like to address the Misc/Questions.
>
> 1. We found that the graph prior method cannot converge within 5k iterations for high-density point clouds (*i.e*., point clouds with more than ~10k points). We further discussed this in L258-L264.
>
> 2. We kindly refer the reviewer to the caption of Table 1. We did not report standard deviations smaller than $1e\text{-}2$ as the scene flow estimation would be close to deterministic.
>
> 3. We ran per scene optimizations during runtime -- similar to Deep Image Prior.
>
>
> **References:**
> 1. Liu, Xingyu, Charles R. Qi, and Leonidas J. Guibas. "FlowNet3D: Learning scene flow in 3d point clouds." Proceedings of the IEEE/CVF Conference on Computer Vision and Pattern Recognition. 2019.
> 2. Pontes, Jhony Kaesemodel, James Hays, and Simon Lucey. "Scene Flow from Point Clouds with or without Learning." 2020 International Conference on 3D Vision (3DV). IEEE, 2020.

---

> > ### Comment · Reviewer_y7UW · 2021-08-30
> > **Response**
> >
> > Thanks for the detailed reply!
> >
> > After the discussion with other reviewers, I'm more convinced about the technical contribution of this work. The additional experimental results in the rebuttal and in the reply to other reviewers are solid and useful, should be added into the future version. Also, the discussion between this work and DeepMapping should be added. I raised my score to accept.

---

### Decision · Program_Chairs · 2021-09-27

**Decision:**

Accept (Spotlight)

**Comment:**

This iinteresting and well written paper proposes to use neural network training at runtime (i.e., during inference) to fit a scene flow function g (a plain MLP) that maps 3D points from a point cloud at time t to time t'. The function is formulated per and applied to individual points as an alternative formulation flow regularisation, meaning that the several thousand points in typical self-driving car dataset are sufficient for it to converge. Strikingly, the method is very competitive to supervised methods (that are faster at runtime but need to be trained) as well as to ICP.

After a lengthy discussion between the 4 reviewers and the authors (longer that the paper itself), the consensus was to accept the paper with scores 6, 7, 7, 7, once the specifics of the evaluations (training the supervised methods on 2k points vs. training on 8k points) were elucidated and the authors accepted to do some revisions to the manuscript. I personally commend all reviewers and authors for exemplary thoroughness and collaboration on this review.